# Electron Capture on Nuclei in Stellar Environment

**Panagiota Giannaka and Theocharis Kosmas ***

Division of Theoretical Physics, University of Ioannina, GR 45100 Ioannina, Greece
* Correspondence: hkosmas@uoi.gr; Tel.: +30-26510-08489

**Abstract:** The stellar electron capture on nuclei is an essential, semi-leptonic process that is especially significant in the central environment of core-collapse supernovae and in the explosive stellar nucleosynthesis. In this article, on the basis of the original (absolute) electron-capture cross-sections under laboratory conditions that we computed in our previous work for a set of medium-weight nuclear isotopes, we extend this study and evaluate folded $e^-$-capture rates in the stellar environment. With this aim, we assume that the parent nuclei and the projectile electrons interact when they are in the deep stellar interior during the late stages of the evolution of massive stars. Under these conditions (high matter densities and high temperatures of the pre-supernova and core-collapse supernova phases), we choose two categories of nuclei; the first includes the $^{48}Ti$ and $^{56}Fe$ isotopes that have $A < 65$ and belong to the iron group of nuclei, and the second includes the heavier and more neutron-rich isotopes $^{66}Zn$ and $^{90}Zr$ (with $A > 65$). In the former, the electron capture takes place mostly during the pre-supernova stage, while the latter occurs during the core-collapse supernova phase. A comparison with previous calculations, which were obtained by using various microscopic nuclear models employed for single-charge exchange nuclear reactions, is also included.

**Keywords:** stellar electron capture; stellar nucleosynthesis; semi-leptonic charged current reactions; quasi-particle RPA

## 1. Introduction

It is well-known that the single-charge exchange process of electron ($e^-$) capture on nuclei, which is represented by the following reaction:

$$(A, Z) + e^- \rightarrow (A, Z-1)^* + \nu_e \tag{1}$$

(where $A$ denotes the mass number and $Z$ the atomic number of the parent nucleus) and takes place in the hot stellar interior [1–4] is one of the essential processes that strongly influences the collapse of the inner core of massive stars, which finally leads to a type II supernova explosion [5–8].

The most significant consequences of this process are the increase of the electron degeneracy pressure, which accelerates the collapse of massive stars, and the enrichment of the nuclear matter composition in the star's interior with neutron-rich isotopes. As a result, large amounts of neutrinos (mostly $\nu_e$) are produced [9–13]. These neutrinos initially have rather low energies and then escape the star (with mass density values of $\varrho < 10^{10}$ g/cm$^3$), carrying away energy and entropy from the core, a process that constitutes an effective cooling mechanism of the exploding massive star. Subsequently, due to electron captures on successively more neutron-rich nuclei and on the free protons, the star's evolution is dominated by de-leptonization (deficit of $e^-$, $e^+$, $\nu_e$, etc.). As soon as the core densities become higher than $\varrho > 10^{11}$ g/cm$^3$, the de-leptonization starts to become blocked, which causes the trapping of the neutrinos that had mostly been produced by the reaction (1) [2,14–16].

From the observations conducted in the last four decades, researchers have concluded that the core of a massive star (progenitor star's mass at $12 < M \leq 20\ M_{solar}$) at the end of its hydrostatic burning is stabilized by the electron degeneracy pressure as long as its core

mass does not exceed the Chandrasekhar mass limit ($M_{Ch}$) [1,3,17,18]. When the core mass exceeds $M_{Ch}$, the electron degeneracy pressure can no longer stabilize the center of the star, and the collapse leads to the initiation of a type II Supernova explosion. Thus, in the early stage of collapse, electrons are captured by nuclei, reducing the electron-to-baryon ratio $Y_e$, while at the same time $\beta$ decay modes become more important and start competing with electron capture [1,11].

In general, the role of electron capture on nucleons and nuclei in the hot interior of massive stars is important because $e^-$-capture drives the evolution of stars, specifically during the last stages of their life, i.e., in the pre-supernova and core-collapse supernova phases. The pertaining conditions deep in the stellar core region in the pre-supernova phase are characterized by high mass densities ($\varrho \leq 10^{10}$ g/cm$^3$) and high temperatures ($T < 0.8$ MeV). Under these conditions, the weak interaction processes (especially the electron capture and the $\beta$-decay modes on nuclei) dominate. Thus, in the pre-supernova phase (but also during the stellar collapse), the core entropy and the ratio $Y_e$ are determined crucially from the above types of electro-weak processes [1,17,18].

Furthermore, the Fermi energy of the degenerate electron gas is sufficiently large compared to the threshold energy $E_{thr}$ (the negative $Q$ value of the reactions involved) in the interior of the stars [19] and leads to appreciable $e^-$-capture on nuclei that reduces the $Y_e$ [8,20]. It is worth noting that the nuclear matter in the stellar core is neutronized, and in this way, the electron pressure is reduced while the energy as well as the entropy decrease. One of the important characteristics of the early pre-explosion evolution is the fact that the role of electron capture is mainly exhibited through the $p - f$ shell nuclei [21,22].

From a nuclear physics point of view, until the early stage of collapse (prevailing mass densities of $\varrho \leq 10^{11}$ g/cm$^3$), electrons are captured on nuclei with a mass number of $45 \leq A \leq 60$–65 [8,10–12,23,24]. Under these conditions (the chemical potential of electrons $\mu_e$ is of the same order of magnitude as the nuclear $Q$ value), the $e^-$ capture cross-sections are sensitive to the details of the Gammow–Teller (GT) strength distributions of the daughter nuclei. Motivated by this effect, many authors emphasized the calculation of $e^-$-capture rates based on the GT-type contributions (at momentum transfer $q \to 0$) and evaluated them on the basis of the dominance of GT transitions [8,10,12,19,22,25]. In our previous work [7], by considering momentum-dependent operators, we saw that in addition to GT transitions, the Fermi (as well as the first- and second-forbidden) transitions hlmay also contribute non-negligible $e^-$-capture rates [8,11,20,22,26].

During the core-collapse phase, the densities ($\varrho \geq 10^{11}$ g/cm$^3$) and the temperatures ($T > 0.8$–1.0 MeV) are high enough and ensure that nuclear statistical equilibrium is achieved. This means that for sufficiently low entropy, the matter composition is dominated by the nuclei with very high binding energy [2]. Under these conditions, $e^-$-capture occurs in more neutron-rich and heavier nuclei, $A \geq 65$ [12,13,20–22,27–29], and as a consequence, the nuclear composition is shifted to more neutron-rich and heavier nuclei [2,3,22].

In this paper, we evaluate $e^-$-capture rates in the stellar environment for the set of isotopes $^{48}Ti$, $^{56}Fe$, $^{66}Zn$, and $^{90}Zr$, which play a prominent role in the pre-supernovae ($^{48}Ti$, $^{56}Fe$) and in core collapse ($^{66}Zn$, $^{90}Zr$) supernovae phases, respectively [15,30,31]. We intend to use the convolution procedure [2,10,11,21,22] to translate the original electron-capture cross-sections of Ref. [7] to those in the stellar environment by assuming that the parent nuclei and the projectile electrons interact deep in the massive stars' interior. The exotic conditions of such an environment favor the consideration of as many low-lying states as possible during the initial state of the parent nucleus. The temperature-dependent energy distribution of these states uses the Maxwell–Boltzmann statistics. On the other hand, the energy distribution of the initial states of electrons are reliably parameterized by the Fermi–Dirac distributions, which depend crucially on the chemical potential of the electron $\mu_e$ [10,11].

We would like to mention that for the calculation of the absolute (original) $e^-$-capture cross-sections in a laboratory environment (conditions) [5,7], our nuclear method (pn-QRPA) provides state-by-state contributions of exclusive, partial, and total $e^-$-capture

rates. The agreement with experimental data [32–39] encouraged us to proceed with the calculations of electron-capture cross-sections in supernova conditions (high densities and high temperatures).

The rest of the article is organized as follows. In Section 2, the theoretical background relevant for the translation of the original cross-sections to those under exotic conditions in the hot stellar interior are briefly described. Then, in Section 3, the partial and total cross-sections for the four isotopes chosen are presented and discussed in detail. Finally, in Section 4, we summarize the main findings extracted from the present study.

## 2. Brief Description of the Theoretical Background

In reaction (1), an electron ($e^-$) of energy $E_e$ is captured by the nucleus $(A, Z)$ interacting weakly with it via $W^\pm$ boson exchange, while the outgoing neutrino $\nu_e$ carries away energy $E_\nu$. The daughter nucleus $(A, Z - 1)$ absorbs a part of the incident electron energy $E$, which (ignoring the nuclear recoil) is given by the difference between the initial and the final nuclear energies as $E = E_f - E_i$ (with $E_i$ and $E_f$ being the energy of the initial and final nuclear states, respectively), and generally appears excited. From the energy conservation in the reaction (1), the energy of the outgoing neutrino $E_\nu$ is written as follows:

$$E_\nu = E_e - Q + E_i - E_f$$

The $Q$-value of the process is determined from the experimental masses of the parent $(M_i)$ and the daughter $(M_f)$ nuclei, expressed as $Q = M_f - M_i$.

As discussed before, reliable stellar simulations in the final collapse and in the explosion phase of massive stars for a plethora of nuclear isotopes throughout the periodic table are required in order to understand the physics of the hot and dense stellar environment. Moreover, since neutrinos—the essential particles in the collapse phase—are mainly produced by $e^-$-capture on nuclei (and on free protons), successful stellar simulations for this phase require accurate neutrino energy spectra that have been generated by the e-capture process (1) [2,11,40].

We note that in general, the neutrino energy spectra emitted through the $e^-$-capture in the star's interior (during the pre-supernova and supernova phases) may be parameterized via an appropriately normalized Fermi–Dirac type of distribution, with parameters such as the chemical potential of the electron $\mu_e$ and the temperature T [22,41–43]. Such energy spectra and $e^-$-capture rates in the stellar environment, however, are limited in the literature [44,45], which is what motivated our present study.

The reaction rates for $e^-$-capture on free protons $\lambda_p$ and on nuclei $\lambda_j$ enter the stellar simulations of core-collapse supernova through the following definitions:

$$R_p = Y_p \lambda_p, \qquad R_N = \sum_j Y_j \lambda_j \equiv Y \cdot \lambda, \tag{2}$$

where $Y_p$ and $Y_j$ represent the number of abundances for free protons and nuclei, respectively. The sum of the latter equation runs over all nuclear isotopes that appear in the stellar core environment. In such calculations, knowledge of the nuclear composition and the $e^-$-capture rates of all nuclear isotopes contained in the stellar core mass are required. Moreover, the rates entering Equation (2) must be known for a wide range of physical parameters, such as the nuclear matter density ($\varrho$) and the temperature ($T$).

The main effort of this type of work is focused on the folded electron-capture rates $\lambda_j$ entering the product $Y \cdot \lambda$ for a set of nuclear isotopes. These rates are written as follows:

$$\lambda_{ec}(T) = \frac{1}{\pi^2 \hbar^3} \int_{E_e^0}^{\infty} \sigma(E_e, T) S_e(E_e, \mu_e, T) \, E_e p_e c \, dE_e \tag{3}$$

where $E_e^0 = max(Q, m_e c^2)$ and $p_e$ is the electron momentum given by the momentum–energy conservation. In the latter equation, $p_e = \sqrt{w^2 - 1}$ represents the electron (positron) momentum, with $w$ being its total energy (rest mass plus kinetic energy); both are in units of

$m_e c^2$. Furthermore, $S_e(E_e, \mu_e, T)$ denotes the Fermi–Dirac $e^-$-distribution (see Appendix A). The quantity $\sigma(E_e, T)$ stands for the total $e^-$-capture cross-section in the stellar environment (see below), while the chemical potential $\mu_e$ is determined as discussed in the Appendix A.

It is worth noting that the rates of the $e^-$-capture process on various nuclear isotopes and the corresponding emitted neutrino spectra in the range of the (T, $\varrho$, $Y_e$) parameters, which describe the star until the core collapse is reached, have been comprehensively studied in Refs. [2,11,40] for a great number of nuclear isotopes with the use of the large-scale shell model. In the present article, we carry out somewhat similar work for the isotopes $^{48}Ti$, $^{56}Fe$, $^{66}Zn$, and $^{90}Zr$ by employing a refined version of the pn-QRPA method [5,7,32] and by performing state-by-state calculations of the stellar $e^-$-capture cross-sections, as stated below.

*Electron-Capture Cross-Sections in the Stellar Environment*

In astrophysical environment, where the finite temperature $T$ and the matter density $\varrho$ effects cannot be ignored, the initial nuclear state needs to be taken as a weighted sum over an appropriate energy distribution. Then, assuming a Maxwell–Boltzmann distribution for the initial state $|i\rangle$ [10,11], the total $e^-$-capture cross-section is given by the following expression [12]:

$$
\begin{aligned}
\sigma(E_e, T) &= \frac{G_F^2 cos^2\theta_c}{2\pi} \sum_i F(Z, E_e) \frac{(2J_i + 1)e^{-E_i/(kT)}}{G(Z, A, T)} \\
&\times \sum_{f,J}(E_e - Q + E_i - E_f)^2 \frac{|\langle i|\widehat{O}_J|f\rangle|^2}{(2J_i + 1)}
\end{aligned} \tag{4}
$$

Thus, the sum over initial states in the latter equation denotes the thermal average of energy levels with the corresponding partition function G(Z,A,T) [12]. In Equation (4), $F(Z, E_e)$ denotes the well-known Fermi function (see Appendix A), and $O_J$ stands for any of the multipole tensor operators (see Appendix of Ref. [5]).

Before elaborating on the specific calculations and the presentation of our results, it is worth mentioning that in calculating the original, total electron-capture cross-sections [7], the use of a quenched value for the static axial-vector coupling constant $g_A$ was necessary for the renormalization of the transition matrix elements [38,39,46,47]. As the coupling constant $g_A$ enters together with the axial-vector form factors $F_A(q^2)$, which multiply the relevant component operators ($\widehat{\mathcal{M}}_{JM}$, $\widehat{\mathcal{L}}_{JM}$, $\widehat{\mathcal{T}}_{JM}^{el}$, and $\widehat{\mathcal{T}}_{JM}^{mag}$) and generate the pronounced excitations $0^-, 1^\pm, \ldots$, etc., the quenched value of $g_A$ obviously influences all these excitations. In fact, in our QRPA calculations, we multiplied the free nucleon coupling constant $g_A = 1.262$ by the factor 0.8 (see Ref. [7] and references therein).

At this point, it is worth mentioning that in Equation (1), in order to measure the excitation energies of the daughter nuclei $(A, Z - 1)$ from the ground state of the parent ones $(A, Z)$, a shifting of the entire set of pn-QRPA states is required [5]. In general, such a shifting is necessary whenever a BCS ground state is used in the pn-QRPA—a treatment previously adopted by other authors [10,48,49]. After the application of the shifting, the resulting low-energy spectrum agrees well with the experimental spectrum of the daughter nucleus. We note that a similar treatment is required in pn-QRPA calculations performed for double-beta decay studies, where the excitations derived for the intermediate odd–odd nucleus (intermediate states) through the p-n and n-p processes from the neighboring nuclei and the left or right nuclear isotope, do not match with each other [48,49].

As discussed before, stellar electron capture plays a crucial role in the late stages of evolution of a massive star, in both the pre-supernova and supernova phases [1,3,17,18]. In the pre-supernova phase, electrons are captured by nuclei with $A \leq 60$–65 [8,10,12,19,22,25], while the collapse-phase electron capture is carried out on heavier and more neutron-rich nuclei, with $Z < 40$ and $N \geq 40$ [12,13,20–22]. The above findings have been taken into account in choosing the set of the nuclear systems studied below.

## 3. Results and Discussion

In this section, we present detailed stellar electron-capture cross-section calculations for the isotopes $^{48}Ti$, $^{56}Fe$, $^{66}Zn$, and $^{90}Zr$ that belong to the medium-weight region of the periodic table. The required nuclear matrix elements between the initial $|J_i\rangle$ and the final $|J_f\rangle$ nuclear states were determined by using the BCS equations for the ground state [32,50,51] and the QRPA equations for the excited states [32,41,51–53]. In the calculation of the matrix elements of the axial vector operators, the quenched value $g_A = 1.00$ was adopted, which subsequently determined all multipole contributions proportional to $g_A$ [38,46,47].

We started with the detailed state-by-state cross-sections of exclusive transitions of the form $|i\rangle \rightarrow |f\rangle$, which are given by the following equation:

$$\left[\frac{d\sigma}{d\omega}\right]^{stel}_{J_f^\pi}(E_e, T, \omega) = \frac{G_F^2 \cos^2\theta_c}{2\pi} \sum_i \frac{e^{-E_i/(kT)}}{G(Z, A, T)} F(Z, E_e)(E_e - Q + E_i - E_f)^2 |\langle i|\widehat{O}_J|J_f^\pi\rangle|^2$$

where $\omega = E_f - E_i$. Then, we calculated the partial contributions of some specific individual multipolarities $J^\pi$. These were obtained by summing over the exclusive contributions of the multipole states of an individual $J^\pi$ multipolarity as follows:

$$\left[\frac{d\sigma}{d\omega}\right]^{stel}_{J^\pi}(E_e, T, \omega) = \sum_f \left[\frac{d\sigma}{d\omega}\right]^{stel}_{J_f^\pi}(E_e, T, \omega)$$

$$= \frac{G_F^2 \cos^2\theta_c}{2\pi} \sum_i F(Z, E_e) \frac{(2J_i + 1)e^{-E_i/(kT)}}{G(Z, A, T)}$$

$$\times \sum_f (E_e - Q + E_i - E_f)^2 \frac{|\langle i|\widehat{O}_J|J_f^\pi\rangle|^2}{(2J_i + 1)} \tag{5}$$

As a specific example of using Equation (5) to obtain partial cross-sections, we evaluate below the contribution of all states of the $J^\pi = 1^+$ multipolarity, which represents the strength of the Gammow–Teller operator.

Finally, we obtained the total stellar cross-sections for a given isotope by summing over the contribution of all accessible multipole states. Practically, this sum only includes the low-spin multipolaries of the daughter nucleus, i.e., those for which $J^\pi \leq 5^\pm$–$6^\pm$. The others contributed negligible portions and were thus ignored. The total stellar cross-sections were obtained as follows:

$$\sigma^{stel}_{tot}(E_e, T) = \sum_{J^\pi} \left[\frac{d\sigma}{d\omega}\right]^{stel}_{J^\pi}(E_e, T, \omega) = \sum_{J^\pi, f} \int \left[\frac{d\sigma}{d\omega}\right]^{stel}_{J_f^\pi}(E_e, T, \omega) \tag{6}$$

We note that in the latter expression for the continuum spectrum of the daughter nucleus, the summation of our state-by-state treatment is practically equivalent to the integration over $\omega$ applied in other methods.

Under the conditions in the stellar interior, where the densities and temperatures are high, for our calculations, we assumed that (i) the initial state of the parent nucleus could be either its ground state or any excited state up to about 3.0 MeV (the contribution of the excited states of the parent nucleus, with energies above 2.5–3.0 MeV, was generally negligible); (ii) the daughter nucleus could be in any accessible final state; (iii) the temperature dependence of the cross-sections could not be ignored (see Section 3.2) [12]; and (iv) all leptons (electrons, positrons, neutrinos, etc.) under stellar conditions had Fermi–Dirac energy distributions (see Appendix A) [10,11].

### 3.1. Stellar e-Capture Rates in Nuclei with $A \leq 65$

In the first stage, our study of the electron-capture process under stellar conditions was restricted to the calculations of cross-sections for two representative examples of the iron group nuclei ($A \approx 45$–65). This was because at pre-supernova conditions, i.e., densities at

$\rho \leq 10^{10}\,\mathrm{g\,cm^{-3}}$ and temperatures at $0.3\,\mathrm{MeV} \leq T \leq 0.8\,\mathrm{MeV}$, electrons were captured by nuclei with $A \leq 60\text{–}65$ [8,10,12,19,22,25]. In this sub-section, we present cross-section calculations of the stellar electron-capture processes that have the $^{48}Ti$ and $^{56}Fe$ isotopes as the parent nuclei.

Due to the fact that the finite temperature induces the thermal population of excited states in the parent nucleus, in obtaining the $e^-$-capture cross-sections as initial states of $^{48}Ti$, we considered the two lowest $0^+$ states, the two lowest $2^+$, and the lowest $4^+$ state. Correspondingly, in the model space chosen, we had 338 accessible final states for the daughter nucleus $^{48}Sc$. Similarly, for the parent nucleus $^{56}Fe$, we assumed that the initial state could be any of the three lowest $2^+$ states, the two lowest $0^+$, and the lowest $4^+$ state, which correspond to 488 excited states of the $^{56}Mn$ daughter nucleus. All of them were involved in the state-by-state calculations performed within our pn-QRPA method.

The results obtained from the study of stellar electron-capture cross-sections for $^{48}Ti$ and $^{56}Fe$ are shown in Figure 1. As can be seen, the general view is similar to that of the original cross-sections of Ref. [7], but now the contributions look higher. In these two figures, it can be observed that with the increase of the mass number A, the threshold for the electron capture changes, which reflects the change in the Q value.

The dominant multipolarity was $1^+$, which contributed more than 40% of the total cross-sections. In the region with energies $E_e \leq 30\,\mathrm{MeV}$, the total $e^-$-capture cross-sections could be well-described only on the basis of the GT transitions, but at higher incident energies $E_e$, the contributions of other multipolarities became remarkable and had to be noted.

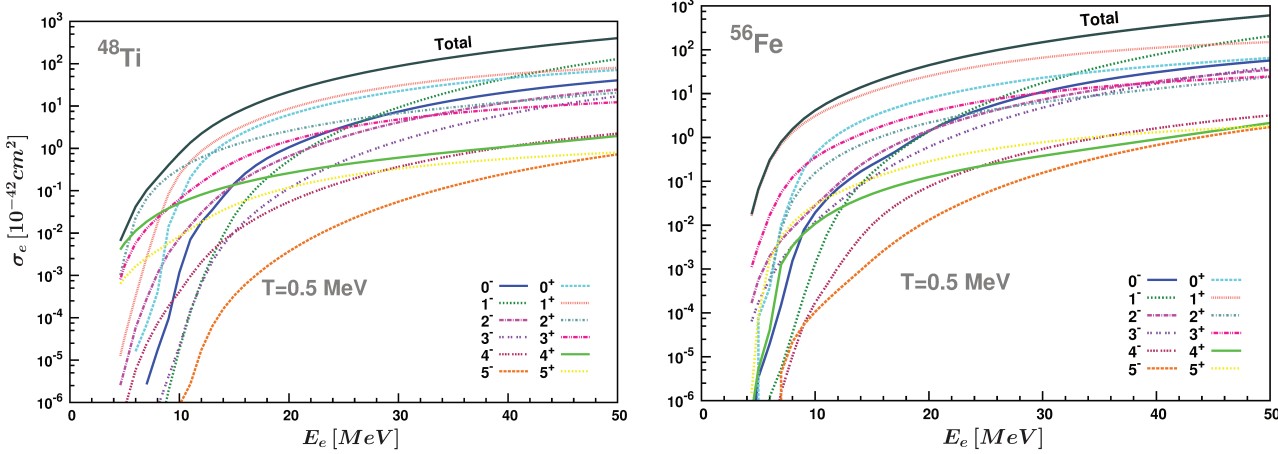

**Figure 1.** Electron-capture cross-sections for the parent nuclei $^{48}Ti$ and $^{56}Fe$ at high temperatures in the stellar environment (T = 0.5 MeV). Total cross-sections and the pronounced individual multipole channels for $J^\pi \leq 5^\pm$ are demonstrated as functions of the incident electron energy $E_e$.

In Table 1, we show the values of partial electron-capture cross-sections on the $^{48}Ti$ nucleus at $T = 0.5\,\mathrm{MeV}$ for different values of the incident electron energy $E_e$ (for $J^\pi \leq 3^\pm$). In Table 2, we tabulate the corresponding cross-sections for the $^{56}Fe$ parent nucleus. After a comparison of these two Tables, we conclude that as the mass number A increases and moves to heavier nuclear isotopes, for a given incident electron energy $E_e$, the partial cross-sections also increase.

**Table 1.** Partial $e^-$-capture cross-sections (in $10^{-42}$ MeV$^{-1}$ cm$^2$) on the $^{48}Ti$ isotope for some representative incident electron energy $E_e$, accessible with different values for the hot stellar interior.

| | $\sigma_e$($\times 10^{-42}$ cm$^2$/MeV) | | | | |
|---|---|---|---|---|---|
| $J^\pi$ | $E_e = 5$ MeV | $E_e = 15$ MeV | $E_e = 25$ MeV | $E_e = 35$ MeV | $E_e = 45$ MeV |
| $0^+$ | 0.00 | 1.795 | 13.233 | 33.438 | 58.169 |
| $1^+$ | $3.53 \times 10^{-5}$ | 2.676 | 18.316 | 43.573 | 69.144 |
| $2^+$ | $2.84 \times 10^{-3}$ | 1.202 | 4.460 | 9.365 | 16.136 |
| $3^+$ | $1.71 \times 10^{-2}$ | 0.477 | 3.015 | 6.765 | 10.545 |
| $0^-$ | 0.00 | 0.146 | 3.251 | 13.545 | 30.476 |
| $1^-$ | 0.00 | 0.255 | 2.833 | 22.606 | 80.689 |
| $2^-$ | $6.61 \times 10^{-5}$ | 0.137 | 1.890 | 7.540 | 17.900 |
| $3^-$ | $1.02 \times 10^{-9}$ | 0.011 | 0.509 | 3.381 | 11.031 |
| Total | $1.13 \times 10^{-2}$ | 6.664 | 48.326 | 142.300 | 298.327 |

**Table 2.** Partial $e^-$-capture cross-sections (in $10^{-42}$ MeV$^{-1}$ cm$^2$) on the $^{56}Fe$ isotope for different incident electron energy $E_e$.

| | $\sigma_e$($\times 10^{-42}$ cm$^2$/MeV) | | | | |
|---|---|---|---|---|---|
| $J^\pi$ | $E_e = 5$ MeV | $E_e = 15$ MeV | $E_e = 25$ MeV | $E_e = 35$ MeV | $E_e = 45$ MeV |
| $0^+$ | $7.75 \times 10^{-5}$ | 3.019 | 14.649 | 32.311 | 53.241 |
| $1^+$ | $6.22 \times 10^{-2}$ | 11.069 | 44.378 | 88.965 | 131.202 |
| $2^+$ | $8.97 \times 10^{-5}$ | 0.872 | 4.025 | 9.253 | 17.589 |
| $3^+$ | $3.38 \times 10^{-3}$ | 1.495 | 6.872 | 14.195 | 21.284 |
| $0^-$ | $3.38 \times 10^{-6}$ | 0.238 | 4.774 | 19.732 | 43.214 |
| $1^-$ | $5.01 \times 10^{-9}$ | 0.156 | 5.988 | 40.144 | 131.845 |
| $2^-$ | $5.67 \times 10^{-4}$ | 0.350 | 3.670 | 12.445 | 26.603 |
| $3^-$ | $1.73 \times 10^{-4}$ | 0.122 | 1.959 | 9.628 | 26.957 |
| Total | $6.64 \times 10^{-2}$ | 17.501 | 87.359 | 229.670 | 458.381 |

*3.2. Stellar e-Capture Rates in Nuclei with A > 65*

During the core-collapse phase (densities at $\rho \geq 10^{10}$ g cm$^{-3}$ and temperatures at $T \simeq 1.0$ MeV), the electron-capture process took place on heavier and more neutron-rich nuclei, with $Z < 40$ and $N \geq 40$ [12,13,20–22]. In this sub-section, cross-section results for the stellar electron capture on the $^{66}Zn$ and $^{90}Zr$ parent nuclei are presented and discussed. Moreover, we study the temperature dependence of the cross-sections on these nuclei. For the $^{66}Zn$ isotope, we could assume that in the stellar environment, its initial state could be either the ground state or a low-lying excited state, with its energy up to about 2.5 MeV. More specifically, we considered the two lowest $0^+$ states, the two lowest $2^+$ states, and the lowest $4^+$ state as initial states, while in the daughter nucleus, many accessible final states could be populated. From the solution of the pn-QRPA equations, in the case of $^{66}Cu$, we found that a total of 447 final states had been included.

As a next system, we chose the $^{90}Zr$ as the parent nucleus, with possible initial states being the two lowest $0^+$, the lowest $2^+$, the lowest $5^-$, and the lowest $3^-$ states. Calculations of the contributions of other states at higher energies for both nuclear isotopes showed that they provided no important contribution to the total e-capture cross-sections. The daughter nucleus in that case was the $^{90}Y$ isotope. In our state-by-state calculations, which we performed in order to obtain the individual contributions to the total cross-sections with $^{90}Zr$ as the parent nucleus, a total of 848 excited states of $^{90}Y$ could be reached.

In Figure 2, where the individual contributions to the total cross-section for $^{66}Zn$ are illustrated, we can see that in addition to the obvious dominant contribution of the $1^+$ multipolarity (found for the other isotopes studied), other multipolarities (such as the $1^-$ and $0^+$) become notable at incident energies $E_e \geq 10$ MeV. For the $^{56}Fe$ isotope, which had an incident energy $E_e$ higher than 42 MeV, the contribution of $1^-$ grew larger than that of

$1^+$. However, the probability of such high $E_e$ energies appearing inside the core plasma is rather small.

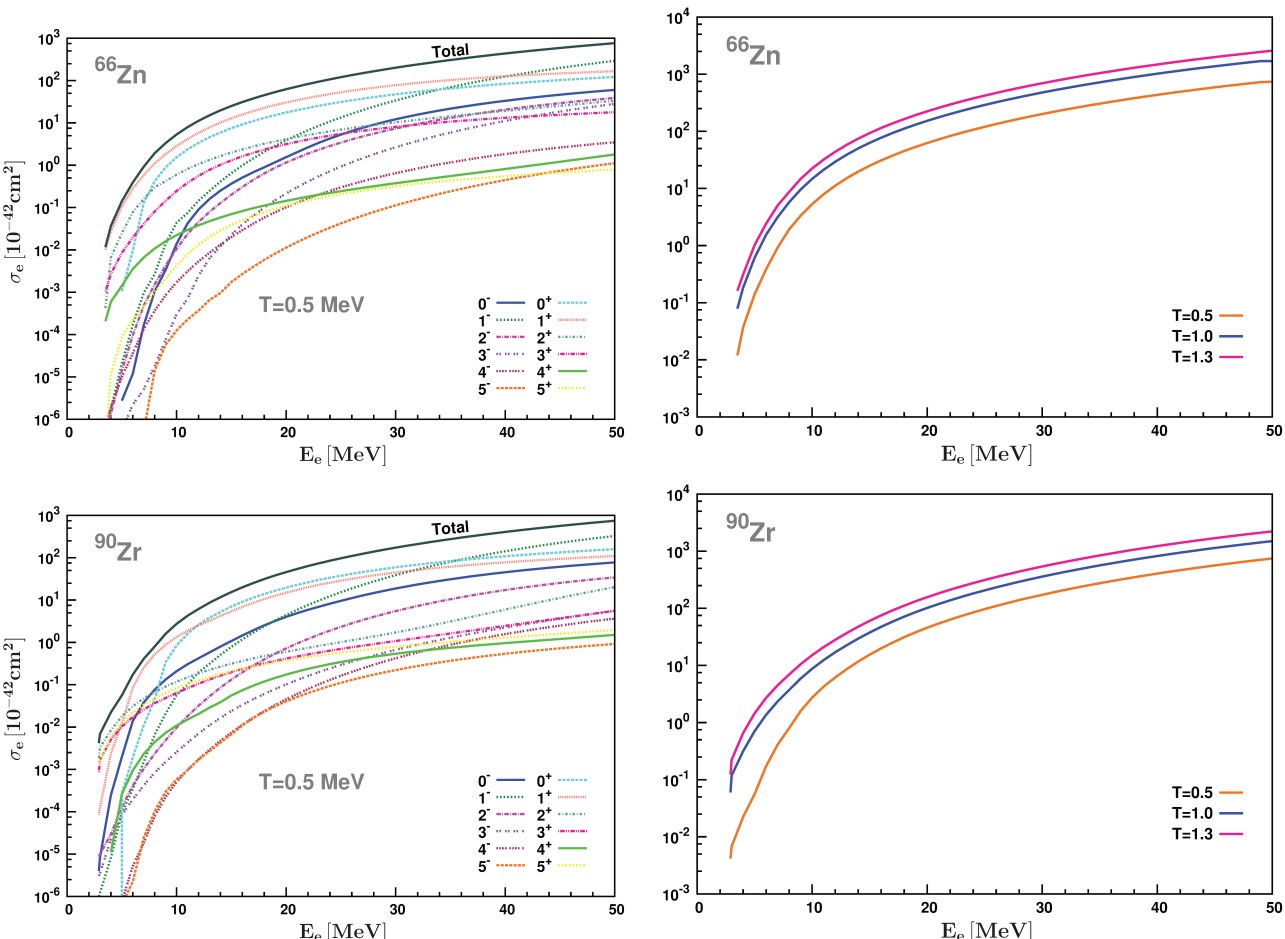

**Figure 2.** The same as Figure 1, but for the parent nuclei $^{66}Zn$ and $^{90}Zr$. Moreover, the right panels show the temperature dependence of the cross-sections on these nuclei.

In the region of energies $E_e \leq 30$ MeV, nearly the entire total $e^-$-capture cross-section may be considered as coming from the GT-type transitions. However, at higher incident energies, the contributions of other multipolarities became notable and could not be omitted. More details, for the values of partial electron-capture cross-sections on the $^{66}Zn$ nucleus at ($T = 0.5$ MeV) for some values of the incident electron energy $E_e$ (for $J^\pi \leq 3^\pm$) obtained by our pn-QRPA method are tabulated in Table 3.

In the study of the heavier isotope $^{90}Zr$, a rather different picture emerged. In this isotope, for low-incident electron energy $E_e$ (up to about 10 MeV), the contribution of the $0^+$ multipolarity was larger than that of the $1^-$, while for $E_e > 30$ MeV, the contribution from the $1^+$ multipolarity became larger than that of $1^-$. It is obvious that in this case, the contribution of this multipolarity must be taken into account.

In Table 4, the values of the partial cross-sections on the $^{90}Zr$ parent nucleus (at temperature T = 0.5 MeV) for various incident electron energies $E_e$ are listed. As can be seen, under these conditions, the $e^-$-capture cross-sections on this heavy nucleus is smaller than that of the $^{66}Zn$.

**Table 3.** Partial $e^-$-capture cross-sections (in $10^{-42}$ MeV$^{-1}$ cm$^2$) on the $^{66}Zn$ isotope for different values of incident electron energy $E_e$.

| | $\sigma_e(\times 10^{-42}$ cm$^2$/MeV) | | | | |
|---|---|---|---|---|---|
| $J^\pi$ | $E_e = 5$ MeV | $E_e = 15$ MeV | $E_e = 25$ MeV | $E_e = 35$ MeV | $E_e = 45$ MeV |
| $0^+$ | $1.06 \times 10^{-3}$ | 7.485 | 31.154 | 64.949 | 102.416 |
| $1^+$ | $1.05 \times 10^{-1}$ | 12.856 | 52.779 | 102.743 | 147.429 |
| $2^+$ | $2.48 \times 10^{-2}$ | 1.986 | 6.922 | 14.306 | 25.297 |
| $3^+$ | $8.53 \times 10^{-3}$ | 1.308 | 5.499 | 10.677 | 15.569 |
| $0^-$ | $2.76 \times 10^{-6}$ | 0.365 | 5.288 | 21.651 | 48.200 |
| $1^-$ | $2.08 \times 10^{-5}$ | 0.649 | 13.409 | 69.666 | 198.475 |
| $2^-$ | $1.40 \times 10^{-5}$ | 0.201 | 3.428 | 13.110 | 29.252 |
| $3^-$ | $3.98 \times 10^{-6}$ | 0.023 | 0.930 | 5.912 | 18.503 |
| Total | $1.41 \times 10^{-1}$ | 25.004 | 120.204 | 305.409 | 588.370 |

**Table 4.** Partial $e^-$-capture cross-sections (in $10^{-42}$ MeV$^{-1}$ cm$^2$) on the $^{90}Zr$ isotope for different values of incident electron energy $E_e$.

| | $\sigma_e(\times 10^{-42}$ cm$^2$/MeV) | | | | |
|---|---|---|---|---|---|
| $J^\pi$ | $E_e = 5$ MeV | $E_e = 15$ MeV | $E_e = 25$ MeV | $E_e = 35$ MeV | $E_e = 45$ MeV |
| $0^+$ | $2.79 \times 10^{-4}$ | 7.186 | 37.775 | 83.375 | 133.684 |
| $1^+$ | $1.23 \times 10^{-2}$ | 5.438 | 28.742 | 61.391 | 93.715 |
| $2^+$ | $1.80 \times 10^{-2}$ | 0.314 | 1.012 | 3.121 | 11.242 |
| $3^+$ | $1.03 \times 10^{-2}$ | 0.198 | 0.701 | 1.623 | 3.719 |
| $0^-$ | $1.97 \times 10^{-3}$ | 1.076 | 9.590 | 30.946 | 61.060 |
| $1^-$ | $9.38 \times 10^{-5}$ | 0.785 | 15.296 | 79.943 | 225.099 |
| $2^-$ | $1.31 \times 10^{-4}$ | 0.132 | 2.329 | 10.461 | 25.351 |
| $3^-$ | $8.39 \times 10^{-5}$ | 0.024 | 0.293 | 1.287 | 3.618 |
| Total | $5.62 \times 10^{-2}$ | 15.431 | 96.929 | 275.166 | 563.511 |

As a final step in our study on the aforementioned set of isotopes, we examined the dependence of the cross-sections on the temperature T prevailing in the stellar interior. In the right panels of the Figure 2, we demonstrate this behavior. We see that as the temperature increases, the total cross-section also increases. For low incident energies, a small change of temperature led to an important increase in the total cross-sections, while for temperatures close to $T = 1.3$ MeV, the total cross-sections were not significantly affected by the increase in temperature.

For both nuclei, above $T = 1.3$ MeV, the total cross-sections remained almost unchanged with increasing temperature. This can be ascribed to the fact that at high temperatures ($T \approx 1.5$ MeV), the GT transitions are thermally unblocked as a result of the excitation of neutrons from the pf-shell into the $g_{9/2}$ orbital, as found by Langanke et al. [54]. Hence, a further increase in temperature did not significantly affect the total cross-sections, implying that at this energy range, the systems could reach the point of saturation.

*3.3. Comparison of Our Rates with Other Model Calculations*

It is worth comparing our present results for the stellar $e^-$-capture cross-sections with those obtained with the use of other nuclear models. Therefore, for this subsection, we chose to compare our folded cross-section for GT transitions with those of Dean et al. [10] and Paar et al. [12]. In the first publication, Dean et al. [10] calculated total electron-capture rates for $^{48}Ti$ and $^{56}Fe$ by using the nuclear shell model and considering only the GT contributions (ignoring the Fermi, first forbidden, second forbidden transitions, etc.). On the other hand, in their calculations, Paar et al. [12] used the relativistic RPA employing a schematic nucleon–nucleon interaction and obtained the total cross-sections by considering contributions from both the Fermi and Gammow–Teller-type operators.

In both the above works, the authors assumed incoming electron energies $E_e$ to lie in the range of $0 \leq E_e \leq 30$ MeV.

In Figure 3, we compare our result for the $1^+$ transitions (G-T transitions) with those obtained in the aforementioned works for stellar temperature $T = 0.5$ MeV. It is very interesting to see that the comparison is good, and that our results agree rather well with those of both previous findings. It should be noted, however, that for the specific value of the axial vector coupling constant $g_A \approx 1.00$ employed in this work (the same for all studied isotopes), throughout the energy range of $0 \leq E_e \leq 30$, our results are a bit higher than both previous results. Hence, better agreement could be achieved for all $E_e$ if we choose a smaller value for $g_A$. Furthermore, the fine structure of this comparison illustrates that our results are in better agreement with those of Paar et al. [12] for $E_e$ energies higher than $E_e \approx 10$ MeV, while for lower energies $E_e \leq 8$ MeV (region of bound states), our results are in better agreement with those of Dean et al. [10]. Finally, from the two isotopes $^{48}Ti$ (left) and $^{56}Fe$ (right), a global picture of all results favors the adopted parameterizations for the three methods in the case of the $^{56}Fe$ isotope.

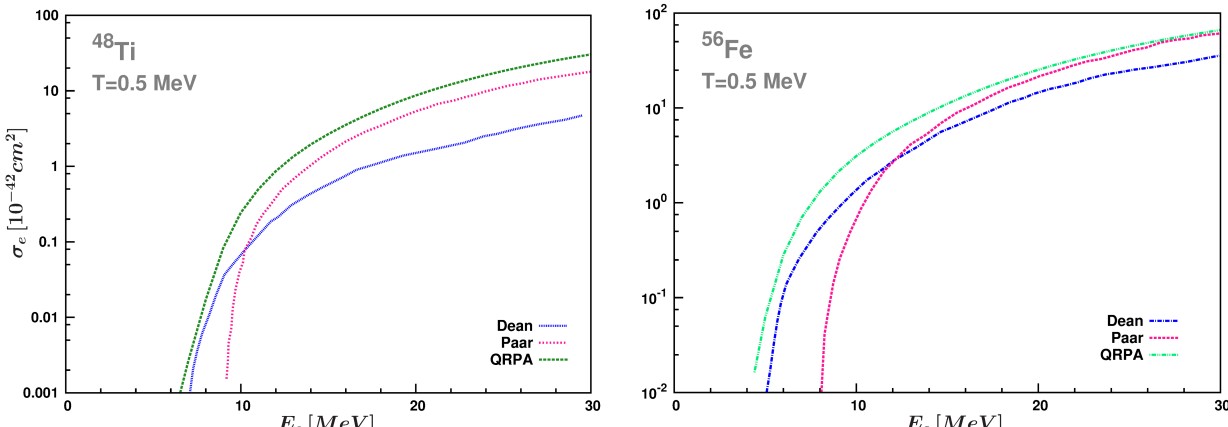

**Figure 3.** Comparison of our GT contribution in total electron-capture rates for $^{48}Ti$ and $^{56}Fe$ with those of other model calculations: (i) Paar et al. [12] and (ii) Dean et al. [10].

Before closing, we should note that in computing the electron capture cross-sections discussed earlier, we did not take into consideration the fact that inside the hot and dense stellar interior, reaction (1) is of the bidirectional type, i.e.,

$$(A, Z) + e^- \rightleftarrows (A, Z - 1)^* + \nu_e. \tag{7}$$

Some authors, in their cross-section calculations, also took into account the reverse channel, namely the charged-current neutrino–nucleus scattering, which in some isotopes may give modified cross-section results. This means that the potential disagreement of our stellar electron-capture cross-sections with those obtained for the reaction (7) would be partially due to the above reason.

## 4. Summary and Conclusions

In stellar evolution and supernova physics, the study of weak interaction processes constitutes a significant topic. Of particular importance is the $e^-$-capture on nuclei, as it plays crucial role in pre-supernova and core-collapse supernova phases as well as in stellar nucleosynthesis. This process predominantly affects the electron-to-baryon ratio $Y_e$ of the matter composition, which leads to more neutron-rich nuclei in the star's interior. The $e^-$-capture on nuclei dominates during the collapse phase, and it becomes increasingly important as the density in the star's central region is enhanced following the increase of the chemical potential of the degenerate electron gas.

By using a numerical approach based on a refinement of the pn-QRPA, which describes several semi-leptonic weak interaction processes well, we performed a detailed study of the electron-capture process on a group of nuclei ($^{48}Ti$, $^{56}Fe$, $^{66}Zn$, and $^{90}Zr$), which are important in the hot and dense stellar environment. We performed state-by-state calculations for the original as well as the stellar cross-sections of $e^-$-capture on the above nuclear isotopes. According to the first conclusions of this study, for incident electron energies $E_e$ up to about 30 MeV, the total $e^-$-capture cross-sections can be reliably calculated by considering only the contribution of the GT transitions, but for higher energies $E_e$ (specifically for heavier and more neutron-rich nuclei), the contribution of other multipolarities are noteworthy and must be taken into account.

Moreover, in our study of the nuclei $^{66}Zn$ and $^{90}Zr$, which play an important role in the collapse phase of a massive star, we found that as the temperature increases up to $T \approx 1.5$ MeV, the total cross-sections also increase. However, a further temperature increase above this value did not significantly affect the total cross-sections, which could have been due to the fact that the unblocking mechanism of GT transitions had already been exhausted. The present calculations are useful in understanding the massive star's evolution in the final stages, the pre-supernova phase, the core-collapse phase, and the supernova explosion if these were to occur.

**Author Contributions:** Both authors equally contributed in all phases of this work. All authors have read and agreed to the published version of the manuscript.

**Funding:** This research was co-financed by the Greek government and the European Union (European Social Fund-ESF) through the Operational Programme "Human Resources Development, Education and Lifelong Learning 2014–2020" in the context of the project (MIS-5047635).

**Institutional Review Board Statement:** Not applicable.

**Informed Consent Statement:** Not applicable.

**Data Availability Statement:** Not applicable.

**Acknowledgments:** P.G. wishes to thank H. Ejiri and T. Shima for their warm hospitality at RCNP, Osaka, Japan during the NNR-19 workshop.

**Conflicts of Interest:** The authors declare no conflicts of interest.

## Abbreviations

The following abbreviations are used in this manuscript:

| | |
|---|---|
| RPA | Random Phase Approximation |
| pn-QRPA | proton–neutron quasi-particle RPA |
| GT | Gamow–Teller |

## Appendix A

*Fermi–Dirac distribution function and chemical potential of $e^{\pm}$:* In the central core stellar environment, the electron (or positron) spectrum is well-described by the known Fermi–Dirac distribution function $S_e$, parameterized with the stellar temperature $T$ and the chemical potential of the electron $\mu_e$ as follows:

$$S_{e,p} = \frac{1}{1 + exp[(E_e - \mu_{e,p})/(k_B T)]} \,. \tag{A1}$$

We note that the positron chemical potential is simply $\mu_p = -\mu_e$, while the Fermi–Dirac distribution for the $e^+$ spectrum results from Equation (A1) by replacing $\mu_e$ with $\mu_p$. In addition, in the core-collapse supernova phase, the neutrinos released through the weak interaction processes that take place in the presence of nuclei (mostly with $45 \leq A \leq 65$) can escape (there is no blocking of neutrinos in the phase space), i.e., $S_\nu \approx 0$.

For the sake of completeness, we mention that in the above case, the connection of the matter density $\varrho$ with the important quantity $Y_e$ (the electron-to-baryon ratio) and the electron (positron) chemical potential $\mu_e$ ($\mu_p$) is written as follows:

$$\varrho Y_e = \frac{1}{\pi^2 N_A} \left(\frac{m_e c}{\hbar}\right)^3 \int_0^\infty (S_e - S_p) p_e^2 dp_e \tag{A2}$$

$S_e$ ($S_p$) is the electron's (positron's) distribution function defined above, and $N_A$ is the well-known Avogadro number; the electron (positron) momentum $p_e$ was defined in Section 2.

*The Fermi function $F(Z, E)$:* The well-known Fermi function employed in this work, $F(Z, E)$, which takes into consideration the final state (Coulomb) interaction of $e^-$, is given in Ref. [55].

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
