# Peer review of "Electron Capture on Nuclei in Stellar Environment"

_2571-712X, doi:10.3390/particles5030030_

Round 1
Reviewer 1 Report
The authors perform calculations of cross sections for electron capture on nuclei under stellar conditions. The performed research is thus relevant e.g. for supernova explosion calculations. The manuscript is thus suitable for publication in Particles. But, before publication I think that the authors should taken into account the following comments:
1) The authors compute the cross sections for electron capture. On the other hand they don't take into account the reverse reaction, i.e. neutrino scattering which could modify the yield. It should at least be commented on why this is neglected.
2) In the calculations they take into account excited states in the initial nucleus. It means that the excited states of the initial nucleus are excitations on a QRPA vacuum |QRPA>. On the other hand, the final states are built on a pnQRPA vacuum |pnQRPA>. These two are general different. The authors should comment on how this is taken into account in their method.
3) On page 10, they compare with calculations of Refs. [10] and [12]. The authors say that they have an excellent agreement with calculations of Paar et al for E_e > 10 MeV. This seems to be an overstatement at least comparing with the left panel of Fig 3. In fact, the authors get cross sections which for most energies overshoots the results of [10] and [12]. They should comment on this. Perhaps, it is related to the chosen value of g_A.
4) On line 339 there is a typo: Firmi -> Fermi
5) I don't see any reason to split Appendix A into two subsections.
Author Response
Manuscript ID: particles-1853846
Title: “Electron Capture on nuclei in stellar environment”, by P. Giannaka, T.S. Kosmas
Reviewer 1: Comments and Suggestions
The authors perform calculations of cross sections for electron capture on nuclei under stellar conditions. The performed research is thus relevant e.g. for supernova explosion calculations. The manuscript is thus suitable for publication in Particles.
Authors’ reply: Yes, exactly, Thank you very much.
Reviewer’s Comment:
But, before publication I think that the authors should taken into account the following comments:
1) The authors compute the cross sections for electron capture. On the other hand they don't take into account the reverse reaction, i.e. neutrino scattering which could modify the yield. It should at least be commented on why this is neglected.
Authors’ reply: We thank the Reviewer for this comment. Towards this end, in the present version we have added (just before the Section “Summary and Conclusions”) a relevant short paragraph.
Reviewer’s Comment:
2) In the calculations they take into account excited states in the initial nucleus. It means that the excited states of the initial nucleus are excitations on a QRPA vacuum |QRPA>. On the other hand, the final states are built on a pnQRPA vacuum |pnQRPA>. These two are general different. The authors should comment on how this is taken into account in their method.
Authors’ reply: The Reviewer is right, the two sets of states are, in general, different and a matching of excitations build on the |pnQRPA> vacuum with the excitation spectra of the daughter nuclei is required. The relevant treatment is discussed in detail in Ref. [5] of the manuscript. For self consistency, however, of the present paper we added a relevant short paragraph in Sect. 2.
Reviewer’s Comment:
3) On page 10, they compare with calculations of Refs. [10] and [12]. The authors say that they have an excellent agreement with calculations of Paar et al for E_e > 10 MeV. This seems to be an overstatement at least comparing with the left panel of Fig 3. In fact, the authors get cross sections which for most energies overshoots the results of [10] and [12]. They should comment on this. Perhaps, it is related to the chosen value of g_A.
Authors’ reply: The Reviewer is right, in this version we rephrased the discussion related to the level of agreement of our results with those of previous calculations.
Reviewer’s minor points:
4) On line 339 there is a typo: Firmi -> Fermi
5) I don't see any reason to split Appendix A into two subsections.
Authors’ reply: Thanks a lot for the careful reading of our manuscript. Both points corrected accordingly.

Reviewer 2 Report
The authors presented a detailed study of the electron capture process in a stellar environment. The research was conducted in a very satisfactory way. The reading is good and no major comments are present. I would suggest the editor accept the paper.
Author Response
Dear Reviewer 2,
The authors wish to thank you very much for the careful reading of our paper. Thank you also for appreciating of our work which gives the good opportunity to appear published in one of the next Volumes of the Journal "Particles"
Thanking you once again,
With our best regards
On behalf of the authors
Haris Kosmas